# Best-Arm Identification Using Extreme Value Theory Estimates of the CVaR

**Dylan Troop** [1,*] , **Frédéric Godin** [2] and **Jia Yuan Yu** [1]

1   Concordia Institute of Information System Engineering, Concordia University,
    Montréal, QC H3G 1M8, Canada; jiayuan.yu@concordia.ca
2   Department of Mathematics and Statistics, Concordia University, Montréal, QC H3G 1M8, Canada;
    frederic.godin@concordia.ca
*   Correspondence: dylantroop@gmail.com

**Abstract:** We consider a risk-aware multi-armed bandit framework with the goal of avoiding catastrophic risk. Such a framework has multiple applications in financial risk management. We introduce a new conditional value-at-risk (CVaR) estimation procedure combining extreme value theory with automated threshold selection by ordered goodness-of-fit tests, and we apply this procedure to a pure exploration best-arm identification problem under a fixed budget. We empirically compare our results with the commonly used sample average estimator of the CVaR, and we show a significant performance improvement when the underlying arm distributions are heavy-tailed.

**Keywords:** sequential decision making; multi-armed bandits; conditional value-at-risk; extreme value theory; heavy-tailed distributions; risk-aware reinforcement learning

## 1. Introduction

In the stochastic multi-armed bandit (MAB) problem, a learning agent is presented with the repeated task of selecting from a number of choices (arms), each providing independent and identically distributed rewards/costs. The agent has no prior knowledge of the underlying arm distributions. Through a combination of exploration and exploitation, the agent attempts to identify the arm with the most favorable reward/cost distribution as quickly as possible; see Lattimore and Szepesvári (2020) for a description of such a setting, which has many applications, such as these detailed in Bouneffouf et al. (2020).

In the traditional MAB framework, the most favorable distribution maximizes the expected reward. Recent generalizations of this problem replace the expectation objective with other metrics aimed at measuring risk. For example, Bhat and Prashanth (2019); David et al. (2018); Galichet et al. (2013); Kagrecha et al. (2019); Sani et al. (2012); Torossian et al. (2019); Yu and Nikolova (2013) addressed the MAB problem with a risk-averse (risk-aware) agent.

Such a framework is better suited to applications in finance where a risk-reward trade-off is sought rather than pure profit maximization. Example of applications of risk-aware multi-armed bandits in finance include portfolio optimization as in Hoffman et al. (2011), Shen et al. (2015), Moeini et al. (2016) and Huo and Fu (2017), cryptocurrency investing as in Yu et al. (2018), hedging as in Cannelli et al. (2020) and energy markets trading as in Costa and Oliveira (2008) and Ariffin et al. (2016).

Often, the risk measure of interest is the *conditional value-at-risk* (CVaR). Given a continuous random variable $X$ representing a cost (i.e., where larger values are less desirable), the CVaR at a confidence level $\alpha \in (0, 1)$ measures the expected value of $X$ given that $X$ exceeds the quantile at level $\alpha$. This quantile is referred to as the *value-at-risk* (VaR). Compared to the VaR, the CVaR captures more information about the weight of a distribution's tail, making it a more useful object of study in risk-averse decision making.

In previous papers on the risk-averse MAB problem, the CVaR is usually estimated by averaging observations above the estimated VaR. When $\alpha$ is close to 1, these observations can be very scarce in small samples, leading to imprecise estimates. This is particularly apparent in *heavy-tailed distributions*, where extreme events correspond to very large values that are not readily observed. This is in contrast to light-tailed distributions, where similar low probability events are closer to the mean and not as catastrophic.

In this paper, we seek to address the CVaR estimation problem described above within the MAB framework, which has applications in any scenario where the primary goal of the agent is to avoid a severely unfavourable outcome, such as financial ruin, patient deaths in clinical trials or the adverse impact of a natural disaster. The MAB problem with heavy-tailed underlying arm distributions has been analyzed in, for example, Bubeck et al. (2013) and Prashanth et al. (2020). As in the latter, we consider the problem of identifying the arm with the lowest CVaR value in a fixed budget (pure exploration) MAB problem.

This paper considers the CVaR for single stages at a time, i.e., not the CVaR on aggregates of rewards across various stages. Instead of estimating the CVaR by sample averaging, we consider an alternative estimator based on extreme value theory (EVT), e.g., (McNeil et al. 2005, Section 7.2.3). EVT has numerous applications in risk management (see for instance Embrechts et al. 1999 and Embrechts et al. 2013), and the present work is an attempt to incorporate its benefits to the MAB problem, reinforcement learning and the machine learning literature.

By selecting a threshold lower than the VaR, it is possible to approximate the tail distribution of a random variable by using a *generalized Pareto distribution* (GPD) and extrapolating beyond the available observations. The major difficulty of this approach is the bias-variance tradeoff that results from threshold selection: too high of a threshold results in limited data availability causing high variance, whereas too low of a threshold can cause a large bias between the GPD and true tail distribution.

Traditionally, threshold selection methods often rely on the visual inspection of graphical plots and apply to a fixed dataset, making them unsuitable for the online learning framework of the MAB problem. Recently, the work of Bader et al. (2018) presented a solution to threshold selection by automating the process using ordered goodness-of-fit tests. This method has recently shown promising experimental results in VaR estimation, i.e., Zhao et al. (2018), and we propose to use this method in an algorithm for the CVaR in MABs.

The main contribution of this paper is the design of a CVaR estimation algorithm that combines the EVT method with automated threshold selection and application of this algorithm in the MAB context. Such an algorithm is crucial (typo) from a practical standpoint since it allows rigorously tackling risk-aware MAB problems requiring avoiding catastrophic risk, a class of problems for which the existing methodologies are ill-adapted due to the scarcity of data related to extreme outcomes. The present paper also bridges the gap between the MAB literature and the extreme value theory one which, to the best of the authors knowledge, have never been previously combined. In our simulation study, this estimation procedure outperformed the commonly used sample average estimator in heavy-tailed distributions.

The remainder of this paper is organized as follows. In Section 2, the VaR and CVaR are formally defined, and the sample average estimator of the CVaR is given. Needed background from EVT is discussed, and we introduce approximate formulas for the VaR and CVaR. We present our CVaR estimation procedure using EVT and automated threshold selection in Section 3, and we introduce the risk-averse successive rejects algorithm for best-arm identification in Section 4. Numerical experiments are presented in Section 5, and we conclude in Section 6.

## 2. Preliminaries

### 2.1. Risk Measures

Let $X$ denote a random variable and $F$ its corresponding cumulative distribution function (CDF). In this paper, we adopt the convention that $X$ represents a cost to the agent, and thus larger values of $X$ are less desirable.

**Definition 1** (Value-at-Risk)**.** *The value-at-risk of X at level $\alpha \in (0,1)$ is*

$$\mathrm{VaR}_\alpha(X) = \inf\{x \in \mathbb{R} \mid F(x) \geq \alpha\}. \tag{1}$$

$\mathrm{VaR}_\alpha(X)$ is equivalent to the quantile at level $\alpha$ of $F$. If the inverse of $F$ exists, $\mathrm{VaR}_\alpha(X) = F^{-1}(\alpha)$. The VaR can be estimated in the same way as the standard empirical quantile. Let $X_1, \ldots, X_n$ be i.i.d. random variables with common CDF $F$. Denote by $X_{(1,n)} \leq X_{(2,n)} \ldots \leq X_{(n,n)}$ the set of order statistics for the sample of size $n$, i.e., the sample sorted in non-decreasing order. An estimator for the VaR is

$$\widehat{\mathrm{VaR}}_{n,\alpha}(X) = \min\left\{ X_{(i,n)} \mid i = 1, \ldots, n; \hat{F}_n\left(X_{(i,n)}\right) \geq \alpha \right\} = X_{(m,n)},$$

where $\hat{F}_n$ denotes the empirical CDF and $m = \lceil \alpha n \rceil$ with $\lceil x \rceil$ denoting the smallest integer greater or equal to $x$. Such an estimate corresponds to the $\alpha$-level VaR of the empirical distribution of $X$ based on the sample of size $n$. We now define the CVaR as in Acerbi and Tasche (2002).

**Definition 2** (Conditional Value-at-Risk)**.** *The conditional value-at-risk of a continuous random variable X at level $\alpha \in (0,1)$ is*

$$\mathrm{CVaR}_\alpha(X) = \mathbb{E}[X \mid X \geq \mathrm{VaR}_\alpha(X)] = \frac{1}{1-\alpha} \int_\alpha^1 \mathrm{VaR}_\gamma(X) d\gamma. \tag{2}$$

Typical values of $\alpha$ are 0.95, 0.99, 0.999, etc. Without loss of generality, the current work only considers continuous random variables. For simplicity of notation, we refer to $\mathrm{CVaR}_\alpha(X)$ and $\mathrm{VaR}_\alpha(X)$ by $c_\alpha$ and $q_\alpha$, respectively, for the remainder of this paper. Typically, the CVaR is estimated by averaging observations above $\widehat{\mathrm{VaR}}_\alpha(X)$. This estimator is given by

$$\widehat{\mathrm{CVaR}}_{n,\alpha}(X) = \frac{\sum_{i=1}^n X_i \mathbb{1}_{\{X_i \geq \widehat{\mathrm{VaR}}_{n,\alpha}(X)\}}}{\sum_{j=1}^n \mathbb{1}_{\{X_j \geq \widehat{\mathrm{VaR}}_{n,\alpha}(X)\}}}, \tag{3}$$

which would coincide with $\mathbb{E}[Z \mid Z \geq \mathrm{VaR}_\alpha(Z)]$ under the assumption that the distribution of $Z$ is the empirical distribution of $X$. The CVaR risk measure has proven extremely popular in the risk management literature due to its favorable theoretical properties. For instance, it is a coherent risk measure as defined by Artzner et al. (1999), since it satisfies the monotonicity, translation invariance, subadditivity and positive homogeneity properties.

The use of Equation (3) can be problematic when the confidence level $\alpha$ is high due to the scarcity of extreme observations. We now provide tools from extreme value theory to address this problem.

### 2.2. Extreme Value Theory and Heavy Tails

Let $F^n(x) = \mathbb{P}(\max(X_1, \ldots, X_n) \leq x)$ denote the CDF of the sample maximum. Suppose there exists a sequence of real-valued constants $a_n > 0$ and $b_n$, $n = 1, 2, \ldots$, and a nondegenerate CDF $H$ such that

$$\lim_{n \to \infty} F^n(a_n x + b_n) = H(x), \tag{4}$$

for all $x$, where *nondegenerate* refers to a distribution not concentrated at a single point. The class of distributions $F$ that satisfy (4) are said to be in the *maximum domain of attraction*

*of H*, denoted $F \in \mathrm{MDA}(H)$. The Fisher–Tippett–Gnedenko theorem (see De Haan and Ferreira (2006, Theorem 1.1.3)) states that $H$ must then be a *generalized extreme value distribution* (GEVD), which is given in the following definition.

**Definition 3** (GEVD). *The generalized extreme value distribution (GEVD) with single parameter* $\xi \in \mathbb{R}$ *has the CDF*

$$H_{\xi}(x) = \begin{cases} \exp\left(-(1+\xi x)^{-1/\xi}\right) & \text{if } \xi \neq 0, \\ \exp(-e^{-x}) & \text{if } \xi = 0, \end{cases}$$

*over its support, which is* $[-1/\xi, \infty)$ *if* $\xi > 0$, $(-\infty, -1/\xi]$ *if* $\xi < 0$ *or* $\mathbb{R}$ *if* $\xi = 0$.

If $F \in \mathrm{MDA}(H)$, then there exists a unique $\xi \in \mathbb{R}$ such that $H = H_{\xi}$. It is important to note that essentially all common continuous distributions used in applications are in $\mathrm{MDA}(H_{\xi})$ for some value of $\xi$, and this value characterizes the speed of the tail decay. When $\xi > 0$, then $F$ is a *heavy-tailed distribution*, which means that moments of order greater than or equal to $1/\xi$ do not exist. The tail decreases like a power law, and very large observations can occur with a non-negligible probability. Otherwise, $F$ is light-tailed with a tail with exponential decay ($\xi = 0$), or the right endpoint of $F$ is finite ($\xi < 0$). If $\xi \geq 1$, then $F$ has infinite mean, and therefore the true CVaR, Equation (2), is also infinite. For the remainder of this paper, we assume that the following condition is satisfied.

**Assumption 1.** $F \in \mathrm{MDA}(H_{\xi})$ *with* $\xi \in [0, 1)$.

When $F \in \mathrm{MDA}(H_{\xi})$, there exists a useful approximation of the distribution of sample extremes above a high threshold.

**Definition 4** (Excess distribution function). *For a given threshold* $u > \mathrm{ess\,inf}\, X$, *the excess distribution function is defined as*

$$F_u(y) = \mathbb{P}(X - u \leq y \mid X > u) = \frac{F(y+u) - F(u)}{1 - F(u)}, \quad y > 0.$$

The $y$-values are referred to as the *threshold excesses*. Given that $X$ has exceeded some high threshold $u$, this function represents the probability that $X$ exceeds the threshold by at most $y$. For large values of $u$, the Pickands–Balkema–de Haan theorem states that $F_u$ can be well-approximated by the GPD. We recall this theorem now.

**Theorem 1** (Balkema and De Haan 1974; Pickands III et al. 1975). *Suppose Assumption 1 is satisfied. Then, there exists a positive function* $\sigma = \sigma(u)$ *such that*

$$\lim_{u \to \infty} \sup_{0 \leq y \leq \infty} |F_u(y) - G_{\xi,\sigma}(y)| = 0, \tag{5}$$

*where* $G_{\xi,\sigma}$ *is the CDF of the generalized Pareto distribution, given by*

$$G_{\xi,\sigma}(y) = \begin{cases} 1 - \left(1 + \frac{\xi y}{\sigma}\right)^{-1/\xi}, & \xi \neq 0, \\ 1 - e^{-y/\sigma}, & \xi = 0, \end{cases} \tag{6}$$

*over its support, which is* $[0, \infty)$ *if* $\xi \geq 0$ *or* $[0, -\sigma/\xi]$ *if* $\xi < 0$.

Using Theorem 1, it is quite straightforward to derive approximate formulas for the VaR and CVaR using the definition of the excess CDF and Equations (1) and (2); for example, see McNeil et al. (2005, Section 7.2.3).

**Corollary 1** (EVT approximations)**.** *Suppose that Assumption 1 is satisfied. Fix $u > $ ess inf $X$ and let $\sigma = \sigma(u)$ be a function satisfying Equation (5). Then, the EVT approximations for the VaR and CVaR at level $\alpha > F(u)$ are given by, respectively,*

$$
q_{u,\alpha} = \begin{cases} u + \frac{\sigma}{\xi}\left(s_{u,\alpha}^{\xi} - 1\right), & \xi \neq 0, \\ u + \sigma \log s_{u,\alpha}, & \xi = 0, \end{cases} \tag{7}
$$

$$
c_{u,\alpha} = \begin{cases} u + \frac{\sigma}{1-\xi}\left(1 + \frac{s_{u,\alpha}^{\xi}-1}{\xi}\right), & \xi \neq 0, \\ u + \sigma(\log s_{u,\alpha} + 1), & \xi = 0, \end{cases} \tag{8}
$$

*where $s_{u,\alpha} = (1 - F(u))/(1 - \alpha)$.*

The accuracy of the EVT approximations depends on how high of a threshold is used. When these approximations are used in statistical estimation, a lower threshold is preferable to make use of as much data as possible; however, this can induce a significant bias. In the next section, we introduce our algorithm for estimating the CVaR with the EVT approximations and automated threshold selection.

## 3. Estimating the CVaR through EVT

### 3.1. Maximum Likelihood Estimation of the GPD Parameters

In this section, we discuss the estimation of $c_{u,\alpha}$ using Corollary 1 and the maximum likelihood. This can be done by first selecting a threshold $u$ and then estimating the GPD parameters using the threshold excesses above $u$. For brevity, we only present the maximum likelihood estimators (MLEs) for $\xi \neq 0$; for the $\xi = 0$ case, see De Haan and Ferreira (2006, Section 3.4).

For an i.i.d. sample $X_1, \ldots X_n$, the threshold excesses are given by $\{X_i - u \mid i = 1, \ldots, n; X_i > u\}$, and we denote them as $Y_1, \ldots, Y_k$. Assuming $k > 0$, the MLEs are obtained by maximizing the approximate log-likelihood function with respect to $\xi$ and $\sigma$,

$$
(\hat{\xi}_u^{(n)}, \hat{\sigma}_u^{(n)}) = \arg\max_{\xi,\sigma} \sum_{i=1}^{k} \log g_{\xi,\sigma}(Y_i), \tag{9}
$$

where $(\xi, \sigma) \in (-1/2, \infty) \times (0, \infty)$ and $g_{\xi,\sigma}$ is the probability density function of the GPD, which, for $\xi \neq 0$, is given by

$$
g_{\xi,\sigma}(y) = \frac{1}{\sigma}\left(1 + \frac{\xi y}{\sigma}\right)^{-1/\xi - 1}.
$$

Based on partial derivatives of the log-pdf with respect to parameters, the resulting maximum likelihood first-order conditions are given by

$$
\frac{1}{k}\sum_{i=1}^{k} \log\left(1 + \frac{\xi Y_i}{\sigma}\right) = \xi, \quad \frac{1}{k}\sum_{i=1}^{k} \frac{Y_i}{\sigma + \xi Y_i} = \frac{1}{\xi + 1}. \tag{10}
$$

A closed-form solution to Equation (10) does not exist; however, the MLEs can be obtained numerically through standard software packages, e.g., `scipy.stats.genpareto` in Python.

Using Equation (8) with the MLEs and an appropriate selection of threshold leads to the following estimator of the CVaR.

**Definition 5** (EVT CVaR estimator). *Suppose that $(\hat{\xi}_u^{(n)}, \hat{\sigma}_u^{(n)})$ are obtained from the excesses above a chosen threshold $u$. Then, for $\alpha > \hat{F}_n(u)$, an estimator for the CVaR is*

$$\hat{c}_{u,\alpha}^{(n)} \equiv u + \frac{\hat{\sigma}_u^{(n)}}{1 - \hat{\xi}_u^{(n)}} \left(1 + \frac{1}{\hat{\xi}_u^{(n)}}\left[\left(\frac{1 - \hat{F}_n(u)}{1 - \alpha}\right)^{\hat{\xi}_u^{(n)}} - 1\right]\right), \tag{11}$$

*with a similar interpretation based on the second line of Equation* (8) *when $\hat{\xi}_u^{(n)} = 0$.*

Note that estimates $(\hat{\xi}_u^{(n)}, \hat{\sigma}_u^{(n)})$ exhibit asymptotic bias as explained in Troop et al. (2021). The estimation procedure from the latter paper allowing to remove asymptotic bias could be explored in the context of multi-bandits in subsequent work.

### 3.2. Choosing the Threshold

The selection of a suitable threshold $u$ is a difficult problem that has been well-studied in the EVT literature. For a survey of approaches for setting the threshold, see Scarrott and MacDonald (2012). Many such approaches involve applying judgment to ultimately select a value of $u$. Typically, sensitivity analyses are performed by altering the threshold values and ensuring results are robust to the choice of $u$. However, a challenging aspect of threshold selection in the machine learning context of the current paper is that $u$ must be decided automatically. We apply the recently developed method of Bader et al. (2018), which uses a combination of ordered goodness-of-fits tests and a stopping rule to choose the optimal threshold. This method provides some assurance that the excesses above the chosen threshold are sufficiently well-approximated the GPD.

The method of Bader et al. (2018) is as follows. Consider a fixed set of thresholds $u_1 < \ldots < u_l$, where, for each $u_i$, we have $k_i$ excesses. The sequence of null hypotheses for each respective test $i$, $i = 1, \ldots, l$, is given by

$$H_0^{(i)}: \quad \text{The distribution of the } k_i \text{ excesses above } u_i \\ \text{follows the GPD.}$$

For each threshold $u_i$, let $\hat{\theta}_i = (\hat{\xi}_{u_i}^{(n)}, \hat{\sigma}_{u_i}^{(n)})$ denote the MLEs computed from the $k_i$ excesses above $u_i$. The Anderson–Darling (AD) test statistic comparing the empirical threshold excesses distribution with the GPD is then calculated. Let $y_{(1)} < \ldots < y_{(k_i)}$ denote the ordered threshold excesses for test $i$ and apply the transformation $z_{(j)} = G_{\hat{\theta}_i}(y_{(j)})$, $j = 1, \ldots k_i$, where $G$ is from Equation (6). The AD statistic for test $i$ is then

$$A_i^2 = -k_i - \frac{1}{k_i}\sum_{j=1}^{k_i}(2j-1)\left[\log\left(z_{(j)}\right) + \log\left(1 - z_{(k_i+1-j)}\right)\right]. \tag{12}$$

Corresponding $p$-values for each test statistic can then be found by referring to a lookup table (e.g., Choulakian and Stephens 2001) or computed on-the-fly. In the present work, such $p$-values are obtained with the `eva` package in R. Using the $p$-values $p_1, \ldots, p_l$ calculated for each test, the ForwardStop rule of G'Sell et al. (2016) was used to choose the threshold. This is done by calculating

$$\hat{w}_F = \max\left\{w \in I \,\middle|\, -\frac{1}{w}\sum_{i=1}^{w}\log(1 - p_i) \leq \gamma\right\}, \tag{13}$$

where $\gamma$ is a chosen significance parameter and $I \subseteq \{1, \ldots, l\}$, $I \neq \emptyset$. Under this rule, the threshold $u_v$ is chosen, where $v = \min\{w \in I \,|\, w > \hat{w}_F\}$. If $\hat{w}_F$ does exist, then no rejection is made and $u_{\min(I)}$ is chosen. If $\hat{w}_F = \max(I)$, then $u_{\max(I)}$ is chosen. The overall CVaR estimation procedure is summarized in Algorithm 1.

**Remark 1.** *In the threshold selection procedure of Bader et al. (2018), $\hat{w}_F$ is given with $I = \{1, \ldots, l\}$; however, we make the modification that $I$ is an arbitrary index set in view of CVaR estimation: since $\hat{c}_{u,\alpha}^{(n)}$ tends to infinity when $\hat{\xi}_u^{(n)}$ tends to 1, in order to ensure reasonable estimates of the CVaR, we use a cutoff parameter $\xi_{max} < 1$, where the MLE $\hat{\xi}_{u_i}^{(n)}$ and corresponding threshold $u_i$ are discarded if $\hat{\xi}_{u_i}^{(n)} > \xi_{max}$.*

**Remark 2.** *Instead of choosing the candidate thresholds $u_1, \ldots, u_l$ directly, it is usually more convenient to choose threshold percentiles $q_1, \ldots, q_l$ and compute $u$ values via the empirical quantile function, i.e., $u_i = \hat{F}_n^{-1}(q_i)$.*

**Remark 3.** *It may happen that Algorithm 1 fails if $I = \varnothing$, in which case no suitable estimates of $\xi$ are found. This is an indication that the underlying data distribution does not satisfy the condition $\xi < 1$ and that the CVaR does not exist. To make Algorithm 1 robust, the sample average estimate is used as a fallback when the latter occurs.*

---

**Algorithm 1:** EVT CVaR estimation with automated threshold selection

---

**Input:** An i.i.d. sample $X_1, .., X_n$. CVaR level $\alpha$, significance parameter $\gamma$, threshold percentiles $0 < q_1, \ldots, q_l < \alpha$, cutoff $\xi_{max} < 1$.

**Output:** $\hat{c}_{u,\alpha}^{(n)}$ if $I \neq \varnothing$, otherwise return the sample average CVaR estimate using Equation (3)

$I \leftarrow \varnothing$
**for** $i = 1, \ldots l$ **do**
  Set $u_i = \hat{F}_n^{-1}(q_i)$
  Compute $(\hat{\xi}_{u_i}^{(n)}, \hat{\sigma}_{u_i}^{(n)})$ from $k_i$ threshold excesses
  **if** $\hat{\xi}_{u_i}^{(n)} \leq \xi_{max}$ **then**
    Compute $A_i^2$ using Equation (12)
    Set $p_i$ to $p$-value for $A_i^2$ using lookup table
    $I \leftarrow I \cup \{i\}$
  **end**
**end**
**if** $I \neq \varnothing$ **then**
  Set $W = \{w \in I \mid -\frac{1}{w} \sum_{i=1}^{w} \log(1 - p_i) \leq \gamma\}$
  **if** $W \neq \varnothing$ **then**
    Compute $\hat{w}_F$ using Equation (13)
    **if** $\hat{w}_F = \max(I)$ **then**
      $v = \max(I)$
    **else**
      $v = \min\{w \in I \mid w > \hat{w}_F\}$
    **end**
    $u = u_v$
  **else**
    $u = u_{\min(I)}$
  **end**
  Compute $\hat{c}_{u,\alpha}^{(n)}$ using Equation (11)
**end**

---

In the next section, we combine Algorithm 1 with an algorithm for best-arm identification in the risk-averse MAB framework.

## 4. Algorithm for Risk-Averse MABs

The MAB framework of this paper involves a finite horizon multi-stage decision setting, where an agent makes decisions at stages $t = 1, \ldots, n$. Let $\{1, \ldots, K\}$ denote a

set of arms, which are possible actions that can be taken at each stage. In the risk-averse setting, we consider the outcome of each draw from an arm to be a cost to the agent. Define the $K$-dimensional random vector $X^t \equiv (X_1^t, \ldots, X_K^t)$, where $X_j^t$ denotes the cost incurred if the arm $j$ is selected at stage $t$. Vectors $X^1, \ldots, X^n$ are assumed to be independent and identically distributed.

Therefore, for all arms $i = 1, \ldots, K$, cost variables $X_i^1, \ldots, X_i^n$ are i.i.d. copies of some random variable $X_i$. Let $\{F_1, \ldots, F_K\}$ denote the respective CDFs of $X_1, \ldots, X_K$; these distribution functions are unknown to the agent. Throughout the rest of the paper, it is assumed that each arm's cost distribution satisfies Assumption 1. The integrability assumption that $\xi_i < 1, i = 1, \ldots, K$ is equivalent to the bounded moment condition often seen when analyzing bandit algorithms under risk criteria in the heavy-tailed domain (e.g., Bhat and Prashanth 2019), which is required for the CVaR to be finite.

We consider the problem of identifying the arm with the *lowest CVaR value* (at a pre-specified value of $\alpha$) in a fixed budget of $n$ stages, where the CVaR is measured on single stage losses instead of on an aggregate of rewards across various stages. The sequence of selected arms is denoted by $a \equiv (a_1, \ldots, a_n)$, where $a_t \in \{1, \ldots, K\}$. When an arm $a_t$ is selected at stage $t$, its associated cost $X_{a_t}^t$ is observed; however, the costs associated with all other arms $\{X_i^t \mid i \neq a_t\}$ remain unobserved. Let $i^* = \arg\min_{i=1,\ldots,K} \mathrm{CVaR}_\alpha(X_i)$, i.e., the arm with the lowest CVaR value. At the end of $n$ stages, the bandit algorithm outputs an arm selection $\hat{i}^* \in \{1, \ldots, K\}$, which is perceived to be optimal by the learning agent.

The best-arm identification algorithm we consider is based on the successive rejects (SR) algorithm of Audibert and Bubeck (2010), modified to select the arm with lowest CVaR value as in Prashanth et al. (2020). The novel aspect of the proposed algorithm is it inclusion of the EVT-based estimation of CVaR. The steps of our original algorithm are presented in Algorithm 2. The algorithm proceeds through $K-1$ phases, eliminating the arm with the highest estimated CVaR value at each phase. The number of samples taken from each arm at each phase is designed such that the total arm samples does not exceed the given fixed budget $n$ at the end of $K-1$ phases. For arms $i = 1, \ldots, K$, we denote CVaR estimates using Algorithm 1 by $\hat{c}_{u,\alpha}^{(m)}(i)$ (at a sample of size $m$).

---

**Algorithm 2:** EVT CVaR-SR algorithm

**Initialize:** $A_1 = \{1, \ldots, K\}$, $\overline{\log}K = \frac{1}{2} + \sum_{i=2}^{K} \frac{1}{i}$, $n_0 = 0$, $n_k = \left\lceil \frac{1}{\overline{\log}K} \frac{n-K}{K+1-k} \right\rceil$.

**Output:** The single unique element in $A_K$

    **for** $k = 1, \ldots K-1$ **do**
        Sample $n_k - n_{k-1}$ costs from each arm in $A_k$.
        Compute CVaR estimates $\hat{c}_{u,\alpha}^{(n_k)}(i)$ for each arm $i$ in $A_k$ using Algorithm 1.
        Set $A_{k+1} = A_k \setminus \arg\max_{i \in A_k} \hat{c}_{u,\alpha}^{(n_k)}(i)$, i.e., remove the arm with the highest CVaR
        estimate (ties broken arbitrarily).
    **end**

---

## 5. Numerical Experiments

In this section, two estimation methods for the CVaR are compared within a simulation study: the sample average estimator of Equation (3) and the EVT estimator of Algorithm 1, which we denote in this section by CVaR-SA and CVaR-EVT, respectively. First, we consider a pure statistical estimation problem where i.i.d. costs from a single arm are sequentially observed, and the cost distribution CVaR estimates are updated at every stage. This allows evaluating the statistical accuracy of both methods. The second experiment embeds the two respective CVaR estimation methods within the MAB problem of Algorithm 2 so as to assess their suitability for sequential action selection. All code to reproduce our results is provided on the following GitHub repository: https://github.com/dtroop/evt-bandits, accessed on 1 April 2022.

*5.1. Single-Arm CVaR Estimation Experiment*

Distributional parameters as in Section The single-arm problem where all costs are i.i.d. samples from an unknown distribution is first considered. The estimation performance of CVaR-SA and CVaR-EVT are compared via the root-mean-square error (RMSE) and absolute bias on five examples from each of the Burr, Fréchet, half-*t*, Lognormal, and Weibull class of distributions.

The Burr, Fréchet, and half-*t* distributions are all truly heavy-tailed in the sense that they satisfy Assumption 1 with $\xi > 0$. The Lognormal and Weibull distributions can be considered borderline between light and heavy-tailed in the sense that they satisfy Assumption 1 with $\xi = 0$; however, their tails can be heavier than exponential depending on their parameters. The distribution function and CVaR formula for each class of distribution is given in the Appendix A.

The experiments consist of performing $N = 1000$ independent runs. Each run consists of sequentially sampling a total of $n = 20{,}000$ independent costs from the single arm, and the CVaR estimates are updated at every interval of 2000 costs. We fix $\alpha = 0.998$ as an example of an extreme risk level. To compute CVaR-EVT, we set $\gamma = 0.1$ and $\xi_{max} = 0.9$ in Algorithm 1, with 20 threshold percentiles $q_1 = 0.79, q_2 = 0.80, \ldots, q_{20} = 0.98$. Plots of the RMSE and the absolute bias from estimating the CVaR across independent runs on simulated data are given in Figures 1 and 2, respectively. Table A1 of Appendix B provides the mean CVaR estimates across runs at the last stage and corresponding true CVaR values. The latter table also contains information related to the percentage of thresholds rejected in the simulations due to $\hat{\xi}_{u_i}^{(n)} > \xi_{max}$, and the average threshold quantile $q_i$ selected in the simulations for each considered distribution.

In terms of RMSE, the performance improvement of CVaR-EVT over CVaR-SA is most apparent in the Burr, Fréchet, and half-*t* distributions, where $\xi > 0$. These distributions have particularly long tails, and so the few samples used in CVaR-SA do not adequately capture a good estimate of the CVaR. The average threshold percentile chosen by Algorithm 1 for these distributions at $n = 20{,}000$ ranges from 0.80 to 0.82, which highlights the fact that CVaR-EVT makes use of a significant portion of data below the 0.998 quantile.

The Lognormal and Weibull distributions have tails that are more distinct from the GPD, and while we still notice improved overall accuracy from using the additional samples in CVaR-EVT, this improvement is less significant than in the truly heavy-tailed cases. CVaR-SA actually outperforms CVaR-EVT in some cases, which occurs most significantly in the Lognormal(1, 1.5) and Weibull(0.5, 1) distributions. The average threshold percentile chosen in these distributions at $n = 20{,}000$ was 0.84 and 0.90, respectively, indicating that the GPD model did not fit well unless a higher threshold is used in CVaR-EVT.

In Table A1, we provide the average threshold selected by Algorithm 1 for all distributions, as well as the average number of thresholds rejected by the cutoff rule (i.e., where $\hat{\xi}_{u_i}^{(n)} > \xi_{max}$). The latter is reported at a sample size of $n = 2000$ since extremely few thresholds were rejected beyond this sample size.

Despite often exhibiting better performance in terms of RMSE, the CVaR-EVT leads to higher bias than the CVaR-SA for several of the simulations, see for instance the two rightmost columns of Figure 2. This is unsurprising since, as mentioned earlier, estimates of the GPD parameters with the maximum likelihood from Equation (9) are known to be asymptotically biased (De Haan and Ferreira 2006). The use of de-biasing procedures for the CVaR-EVT estimates, such as that of Troop et al. (2021) could therefore be contemplated to further improve the estimation performance.

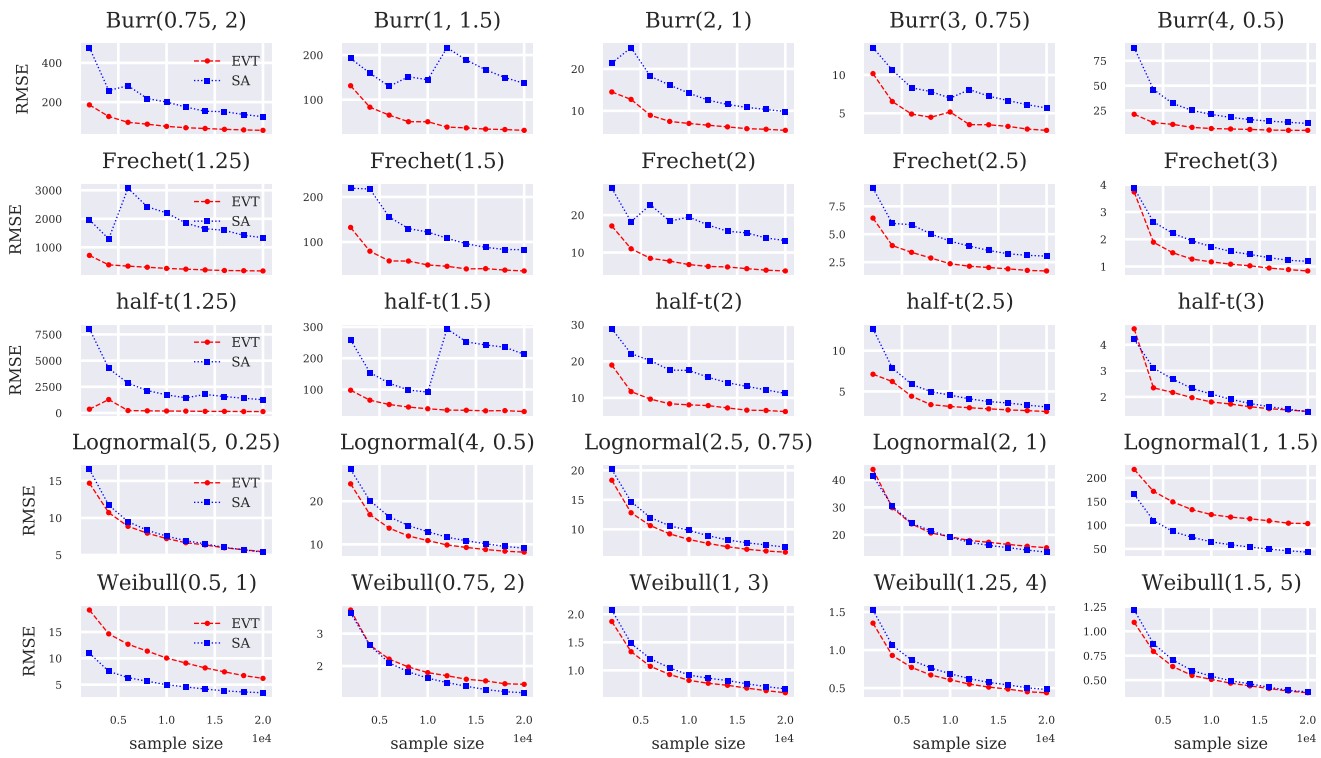

**Figure 1.** RMSE of CVaR-EVT and CVaR-SA with $\alpha = 0.998$.

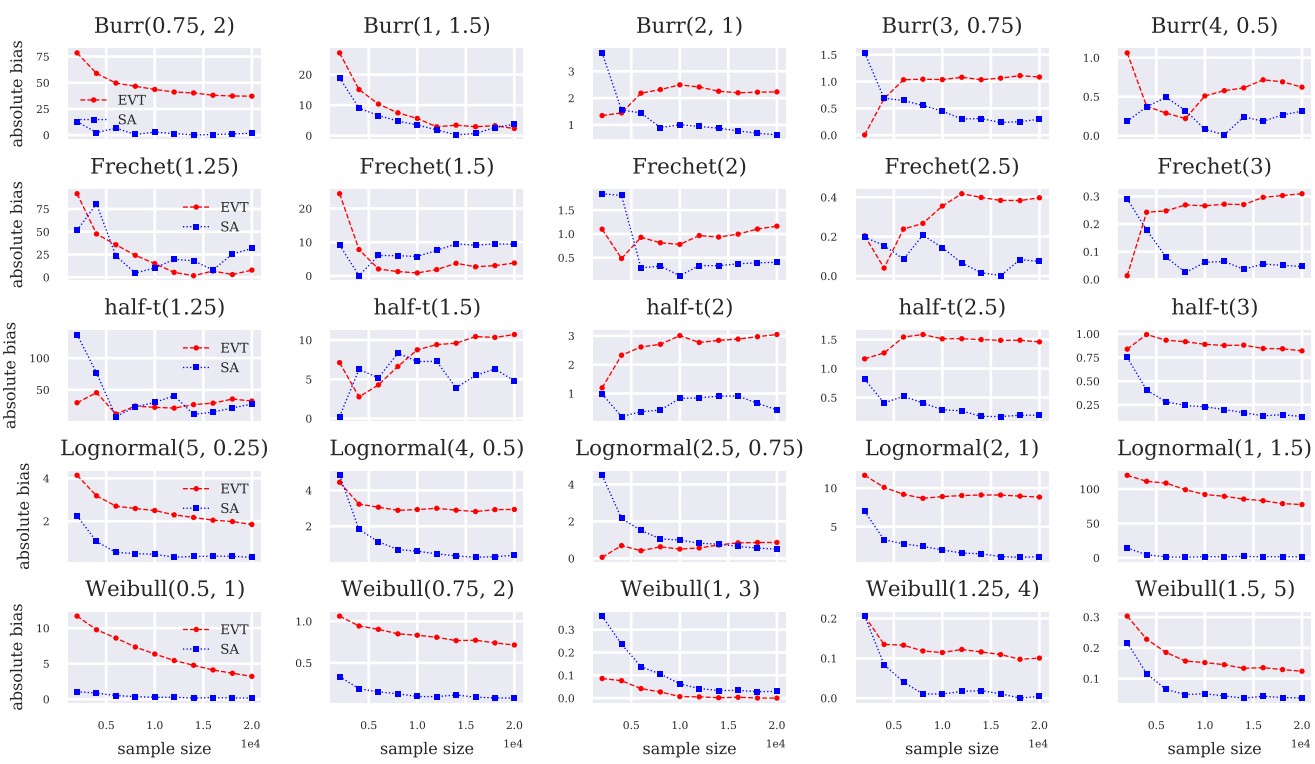

**Figure 2.** Absolute bias of CVaR-EVT and CVaR-SA with $\alpha = 0.998$.

## 5.2. Multi-Armed Bandit Experiment

In this section, we compare the performance of CVaR-EVT and CVaR-SA in a best-arm identification problem. Algorithm 2 is compared with Prashanth et al. (2020, Algorithm 1)

(i.e., where CVaR estimates are computed using CVaR-SA) in five five-arm MAB settings, where in each setting, the five underlying arm distributions come from the same distribution class, either the Burr, Fréchet, half-*t*, Lognormal or Weibull. Again, we fix $\alpha = 0.998$ and use the same distributional parameters as in Section 5.1. In other words, each of the five arms in the bandits experiment has one of the five parameter sets presented in the first experiment. Such parameters were chosen to vary the shape of the underlying distributions while keeping the arm CVaRs relatively close in magnitude, making the task of distinguishing them more difficult to the agent. Both Algorithm 2 and Prashanth et al. (2020, Algorithm 1) are executed at fixed budgets of $n = 5000, 10,000, \ldots, 25,000$, and $N = 1000$ independent runs are performed at each budget. The performance metric we use to compare the two methods is the probability of incorrect best-arm identification, i.e., $\mathbb{P}(\hat{i}^* \neq i^*)$, estimated empirically at a given fixed budget by

$$\frac{\sum_{j=1}^N \mathbb{1}\{\hat{i}_j^* \neq i^*\}}{N},$$

where $\hat{i}_j^*$ denotes the estimated optimal arm at the considered stage for independent run $j, j = 1, \ldots, N$, and $i^*$ denotes the arm with the lowest CVaR value. Plots of the empirical estimates of the probability of incorrect best-arm identification using CVaR-EVT and CVaR-SA are given in Figure 3. Consistently with the results of Section 5.1, the improved accuracy of CVaR-EVT in these classes of distributions leads to a lower error rate in the MAB setting for all budgets in our experiments compared to CVaR-SA.

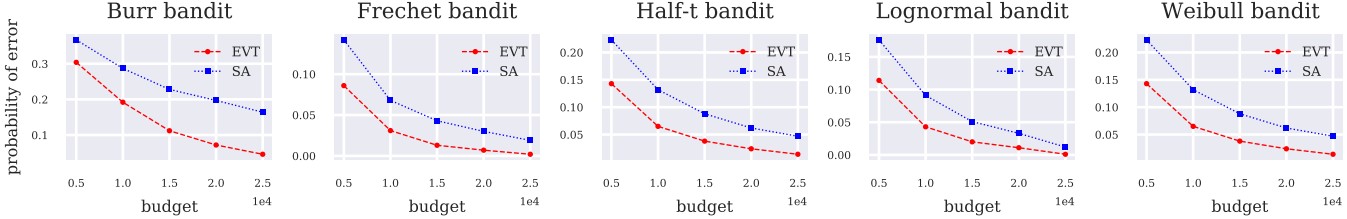

**Figure 3.** The empirical probability of incorrect best-arm identification using CVaR-EVT and CVaR-SA in the successive rejects algorithm.

## 6. Conclusions

We developed a new algorithm for CVaR estimation combining EVT and automated threshold selection and used it in a risk-averse, best-arm identification MAB problem. While the CVaR approximation of Equation (8) exists in the literature, its efficacy in statistical estimation is limited by the problem of threshold selection, which can be unreliable in practice.

The novelty of our approach from a computational perspective is to integrate the sequential goodness-of-fit test of Bader et al. (2018) to develop an automated CVaR estimation algorithm using EVT in view of sequential decision making based on extreme risk. We demonstrated empirically that our algorithm produced more accurate estimates compared to the more commonly encountered sample average CVaR estimator in certain distributions, and is a preferable choice in the risk-averse MAB setting when the underlying arm distributions are heavy-tailed and the confidence level of the CVaR $\alpha$ is large, i.e., close to one.

The present paper has implications from both the academic and the practitioners' perspective. First, our work is a useful addition to the risk-aware reinforcement learning strand of literature, where the key differentiating feature of the present paper is the focus on extreme risk. Moreover, our results also have implications for practitioners as our algorithm can readily be deployed by risk managers to optimize the exposure to catastrophic risk in the context of sequential decision making, for instance in financial contexts.

Among future potential work, testing the proposed algorithm on real data would be of interest. Indeed, while the present study establishes that our algorithm provides good performance in a controlled environment with simulated heavy-tailed data, confirming that such favorable performance still persists in a realistic context is important. Furthermore, as a potential extension, the de-biasing procedure for EVT CVaR estimates from Troop et al. (2021) could be integrated into the multi-armed bandits selection algorithm to further improve the results. Finally, considering alternative objective functions instead of the CVaR could also be explored. For instance, despite being harder to interpret, expectiles are a popular choice in the risk management literature due to their favorable properties, such as elicitability; see for instance Daouia et al. (2018). The extreme-value-based estimation of expectiles could thus also be integrated into a risk-aware MAB framework.

**Author Contributions:** Conceptualization, methodology and writing: D.T., F.G. and J.Y.Y.; programming and software implementation: D.T. All authors have read and agreed to the published version of the manuscript.

**Funding:** Financial support from Natural Sciences and Engineering Research Council of Canada (NSERC, Godin: RGPIN-2017-06837, Yu: RGPIN-2018-05096) is gratefully acknowledged.

**Conflicts of Interest:** The authors declare no conflict of interest.

## Appendix A. Probability Distributions and Their CVaRs

### Appendix A.1. Burr

The Burr distribution with parameters $c, d$ has CDF given by

$$F_{c,d}(x) = 1 - (1 + x^c)^{-d}, \quad c, d, x > 0.$$

The CVaR for the Burr distribution can be derived from its expression for the conditional moment given in Kumar (2017, Section 2.2). If $X \sim \text{Burr}(c, d)$,

$$\text{CVaR}_\alpha(X) = \frac{d[(1/q_\alpha)^c]^{d-1/c}}{(1-\alpha)(d-1/c)} \, {}_2F_1\left(d - \frac{1}{c}, 1 + d, d - \frac{1}{c} + 1; -\frac{1}{q_\alpha}\right), \quad cd > 1, \qquad \text{(A1)}$$

where ${}_2F_1$ denotes the hypergeometric function and $q_\alpha = F_{c,d}^{-1}(\alpha)$. The Burr distribution is in $\text{MDA}(H_\xi)$ with $\xi = 1/(cd)$.

### Appendix A.2. Fréchet

The Fréchet distribution with parameter $\gamma$ has CDF given by

$$F_\gamma(x) = e^{-x^{-\gamma}}, \quad \gamma, x > 0.$$

The CVaR for the Fréchet distribution can be derived from Equation (2). If $X \sim \text{Fréchet}(\gamma)$,

$$\text{CVaR}_\alpha(X) = (1-\alpha)^{-1}[\Gamma(\gamma - 1/\gamma) - \Gamma(\gamma - 1/\gamma, -\log(\alpha))], \quad \gamma > 1, \qquad \text{(A2)}$$

where $\Gamma(\cdot)$ and $\Gamma(\cdot, \cdot)$ denote the gamma and upper incomplete gamma functions, respectively. The Fréchet distribution is in $\text{MDA}(H_\xi)$ with $\xi = 1/\gamma$.

### Appendix A.3. Half-t

If $X$ follows the $t$ distribution with $\nu$ degrees of freedom, then $|X|$ follows the half-$t$ distribution, which has a CDF given by

$$F_\nu(x) = 2 - \mathcal{I}_{t(x)}\left(\frac{\nu}{2}, \frac{1}{2}\right), \quad \nu > 0, x \geq 0,$$

where $t(x) = \frac{\nu}{x^2 + \nu}$ and $\mathcal{I}_t(a, b)$ is the regularized incomplete Beta function. The CVaR for the half-$t$ distribution can be derived from the expression for the CVaR of the $t$-distribution given in (Norton et al. 2019, Proposition 12). If $X \sim$ half-$t(\nu)$, then

$$\text{CVaR}_\alpha(X) = 2 \frac{\nu + q_\alpha}{(\nu - 1)(1 - \alpha)} g_\nu(q_\alpha), \quad \nu > 1,$$

where $g_\nu$ is the probability density function of the standardized $t$-distribution, and $q_\alpha = T^{-1}\left(\frac{\alpha + 1}{2}\right)$ where $T^{-1}$ is the inverse of the CDF of standardized $t$-distribution. The half-$t$ distribution is in MDA($H_\xi$) with $\xi = 1/\nu$.

*Appendix A.4. Lognormal*

The Lognormal distribution with parameters $\mu, \sigma$ has a CDF given by

$$F_{\mu,\sigma}(x) = \frac{1}{2} + \frac{1}{2} \text{erf}\left[\frac{\ln x - \mu}{\sqrt{2}\sigma}\right], \quad \sigma, x > 0.$$

The CVaR for the Lognormal distribution is given in (Norton et al. 2019, Proposition 9). If $X \sim$ Lognormal($\mu, \sigma$),

$$\text{CVaR}_\alpha(X) = \frac{e^{\mu + \sigma^2/2}}{1 - \alpha} \Phi\left[\sigma - \frac{\Phi^{-1}(\alpha)}{\sqrt{2}}\right],$$

where $\Phi$ and $\Phi^{-1}$ are, respectively, the standard normal CDF and its inverse. The Lognormal distribution is in MDA($H_\xi$) with $\xi = 0$.

*Appendix A.5. Weibull*

The Weibull distribution with parameters $\kappa, \lambda$ has a CDF given by

$$F_{\kappa,\lambda}(x) = 1 - e^{-(x/\lambda)^\kappa}, \quad \kappa, \lambda > 0, x \geq 0.$$

The CVaR for the Weibull distribution is given in (Norton et al. 2019, Proposition 13). If $X \sim$ Weibull($\kappa, \lambda$),

$$\text{CVaR}_\alpha(X) = \frac{\lambda}{1 - \alpha} \Gamma\left(1 + \frac{1}{\kappa}, -\log(1 - \alpha)\right),$$

where $\Gamma(a, b) = \int_b^\infty p^{a-1} e^{-p} \, dp$ is the upper incomplete gamma function. The Weibull distribution is in MDA($H_\xi$) with $\xi = 0$.

## Appendix B. Numerical Results

**Table A1.** Data for all distributions used in the experiments. $CVaR_\alpha$ denotes the exact CVaR value for $\alpha = 0.998$. Given at a sample size of $n = 20{,}000$, CVaR-SA and CVaR-EVT denote the average estimated CVaR values across $N = 1000$ independent runs, and threshold pct. denotes the average threshold percentile chosen by Algorithm 1. The rejection rate denotes the average portion of candidate thresholds rejected by the cutoff rule in Algorithm 1, i.e., the total number of times where $\hat{\xi}_{u_i}^{(n)} > \xi_{max}$ divided by the total number of thresholds tested (i.e., a proportion over all runs and thresholds). This value is given at a sample size of $n = 2000$ since extremely few thresholds were rejected beyond this sample size.

| | $CVaR_\alpha$ | CVaR-SA | CVaR-EVT | Threshold Pct. | Rejection Rate |
|---|---|---|---|---|---|
| Burr(0.75, 2) | 184.5 | 182.79 | 221.65 | 0.8 | 0.05 |
| Burr(1, 1.5) | 187.99 | 184.29 | 190.3 | 0.8 | 0.03 |
| Burr(2, 1) | 44.71 | 44.08 | 42.47 | 0.8 | 0.0 |
| Burr(3, 0.75) | 28.5 | 28.2 | 27.41 | 0.8 | 0.0 |
| Burr(4, 0.5) | 44.72 | 44.41 | 44.1 | 0.8 | 0.0 |
| Frechet(1.25) | 721.25 | 689.63 | 713.4 | 0.8 | 0.18 |
| Frechet(1.5) | 188.96 | 179.57 | 185.14 | 0.8 | 0.03 |
| Frechet(2) | 44.71 | 45.12 | 43.55 | 0.8 | 0.0 |
| Frechet(2.5) | 20.02 | 20.09 | 19.62 | 0.8 | 0.0 |
| Frechet(3) | 11.9 | 11.86 | 11.59 | 0.8 | 0.0 |
| half-t(1.25) | 530.66 | 503.19 | 498.46 | 0.8 | 0.17 |
| half-t(1.5) | 156.58 | 161.38 | 145.87 | 0.8 | 0.03 |
| half-t(2) | 44.7 | 45.13 | 41.65 | 0.81 | 0.0 |
| half-t(2.5) | 23.1 | 22.91 | 21.64 | 0.82 | 0.0 |
| half-t(3) | 15.41 | 15.28 | 14.59 | 0.82 | 0.0 |
| Lognormal(5, 0.25) | 328.63 | 328.32 | 326.79 | 0.84 | 0.0 |
| Lognormal(4, 0.5) | 269.11 | 268.73 | 266.18 | 0.8 | 0.0 |
| Lognormal(2.5, 0.75) | 134.45 | 133.96 | 135.32 | 0.8 | 0.0 |
| Lognormal(2, 1) | 183.83 | 182.69 | 192.67 | 0.8 | 0.0 |
| Lognormal(1, 1.5) | 351.98 | 350.52 | 429.49 | 0.84 | 0.0 |
| Weibull(0.5, 1) | 53.05 | 52.9 | 56.26 | 0.9 | 0.0 |
| Weibull(0.75, 2) | 27.99 | 27.92 | 28.71 | 0.82 | 0.0 |
| Weibull(1, 3) | 21.64 | 21.61 | 21.64 | 0.8 | 0.0 |
| Weibull(1.25, 4) | 19.41 | 19.41 | 19.31 | 0.81 | 0.0 |
| Weibull(1.5, 5) | 18.63 | 18.6 | 18.51 | 0.82 | 0.0 |

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
