# Peer review of "Best-Arm Identification Using Extreme Value Theory Estimates of the CVaR"

_jrfm, doi:10.3390/jrfm15040172_

Round 1
Reviewer 1 Report
Dear Authors,
Thank you very much for your submission, i really enjoyed reading it. I really believe that the Reinforcement Learning framework provides real opportunities to address risk measurment issues. I would merely recommend:
1 - Add appropriate reference on the topic (Paper and books from Guégan et al. or Embrechts, etc. seems rather interesting.)
2 - Based on the alternative papers for threshold selection, for CVaR Calculations etc. I would recommend analysing alternative methodologies and potential benchmarks, as though your paper is scientifically sound, simpler approaches might do the job.
3 - I would recommend improving the description of the added value of the MAB approach from a risk management point of view, in the conclusion for instance.
Best Regards,
Your reviewer
Author Response
See the pdf (part Reviewer 1)

Reviewer 2 Report
The research was done carefully. The results seems pretty interesting.
Author Response
N/A, the reviewer did not ask for modifications
Reviewer 3 Report
The authors consider a risk-aware multi-armed bandit framework with the goal of avoiding catastrophic risk. Such a framework has multiple applications in financial risk management. They introduce a new conditional value-at-risk (CVaR) estimation procedure combining extreme value theory with automated threshold selection by ordered goodness-of-fit tests, and apply this procedure to a pure exploration best-arm identification problem under a fixed budget. The authors claim that they have compared empirically with the commonly used sample average estimator of the CVaR, and showed a significant performance improvement when the underlying arm distributions are heavy-tailed.
I find the findings are interesting. I have the following comments for the authors to improve their paper:
- The authors should state the motives of their study and tell readers why their study is important and useful to academics and practitioners. They should also state clearly their contributions to the literature in the introduction section.
- The authors should include a literature review section to discuss all important works and pioneer works and some of the recent studies related to their study
- Most of the theories discussed in Sections 2 and 3 are well known. The authors should cite at least one paper for each well-known result/definition and state clearly which results are derived by the authors.
- Section 4 is too short. They should discuss the algorithm more. Is the algorithm proposed by the authors or by others? They should mention it clearly.
- Section 5 is too brief. The authors discuss the results more. They should also get real data to demonstrate their approach is superior.
- The authors claim that they have compared empirically with the commonly used sample average estimator of the CVaR, and showed a significant performance improvement when the underlying arm distributions are heavy-tailed. They should discuss this clearly. The authors may consider to conduct simulation to show that their approach is better.
- The conclusion section is too brief. They should discuss clearly the motives of their study, tell readers why their study is important and useful to academics and practitioners, state clearly their contributions to the literature in the conclusion section. They should point out the limitations of their approach and suggest directions for further study.
Author Response
See the pdf document, section Reviewer 3

Round 2
Reviewer 1 Report
Dear Authors,
as all points have been addressed, i deem the paper suitable for publication.
best,
bertrand
Reviewer 3 Report
I find the model they developed and their illustration are interesting and useful. The paper is well written. Thus, I recommend accepting their paper.